# DISENTANGLED CONDITIONAL VARIATIONAL AUTOENCODER FOR UNSUPERVISED ANOMALY DETECTION

## ABSTRACT

Recently, generative models have shown promising performance in anomaly detection tasks. Specifically, autoencoders learn representations of high-dimensional data, and their reconstruction ability can be used to assess whether a new instance is likely to be anomalous. However, the primary challenge of unsupervised anomaly detection (UAD) is in learning appropriate disentangled features and avoiding information loss, while incorporating known sources of variation to improve the reconstruction. In this paper, we propose a novel architecture of generative autoencoder by combining the frameworks of $\beta$-VAE, conditional variational autoencoder (CVAE), and the principle of total correlation (TC). We show that our architecture improves the disentanglement of latent features, optimizes TC loss more efficiently, and improves the ability to detect anomalies in an unsupervised manner with respect to high-dimensional instances, such as in imaging datasets. Through both qualitative and quantitative experiments on several benchmark datasets, we demonstrate that our proposed method excels in terms of both anomaly detection and capturing disentangled features. Our analysis underlines the importance of learning disentangled features for UAD tasks.

## 1 INTRODUCTION

Unsupervised anomaly detection (UAD) has been a fertile ground for methodological research for several decades. Recently, generative models, such as Variational Autoencoders (VAEs) (Kingma & Welling, 2014) and Generative Adversarial Networks (GANs) (Goodfellow et al., 2020; Arjovsky et al., 2017), have shown exceptional performance at UAD tasks. By learning the distribution of normal data, generative models can naturally score new data as anomalous based on how well they can be reconstructed. For a recent review of deep learning for anomaly detection, see Pang et al. (2021).

In a complex task like UAD, disentanglement as a meta-prior encourages latent factors to be captured by different independent variables in the low-dimensional representation. This phenomenon has been on display in recent work that has used representation learning as a backbone for developing new VAE architectures. Some of the methods proposed new objective functions (Higgins et al., 2017; Mathieu et al., 2019), efficient decomposition of the evidence lower bound (ELBO) (Chen et al., 2018), partitioning of the latent space by adding a regularization term to the mutual information function (Zhao et al., 2017), introducing disentanglement metrics (Kim & Mnih, 2018), and penalizing total correlation (TC) loss (Gao et al., 2019). Penalized TC efficiently learns disentangled features and minimizes the dependence across the dimension of the latent space. However, it often leads to a loss of information, which leads to lower reconstruction quality. For example, methods such as $\beta$-VAE, Disentangling by Factorising (FactorVAE) (Kim & Mnih, 2018), and Relevance FactorVAE (RFVAE) (Kim et al., 2019) encourage more factorized representations with the cost of either losing reconstruction quality or losing a considerable among of information about the data and drop in disentanglement performance. To draw clear boundaries between an anomalous sample and a normal sample, we must minimize information loss.

To address these limitations, we present Disentangled Conditional Variational Autoencoder (dC-VAE). Our approach is based on multivariate mutual information theory. Our main contribution is

a generative modeling architecture which learns disentangled representations of the data while minimizing the loss of information and thus maintaining good reconstruction capabilities. We achieve this by modeling known sources of variation, in a similar fashion as Conditional VAE (Pol et al., 2019).

Our paper is structured as follows. We first briefly discuss related methods (Section 2), draw connection between them, and present our proposed method dCVAE (Section 3). In Section 4, we discuss our experimental design including competing methods, datasets, and model configuration. Finally, experimental results are presented in Section 5, and Section 6 concludes this paper.

## 2 RELATED WORK

In this section, we discuss related work on autoencoders. We focus on two types of architecture: extensions of VAE enforcing disentanglement, and architectures based on mutual information theory.

### 2.1 $\beta$-VAE

$\beta$-VAE and its extensions proposed by (Higgins et al., 2017; Mathieu et al., 2019; Chen et al., 2018) is an augmentation of the original VAE with learning constraints of $\beta$ applied to the objective function of the VAE. The idea of including such a hyper-parameter is to balance the latent channel capacity and improve the reconstruction accuracy. As a result, $\beta$-VAE is capable of discovering the disentangled latent factors and generating more realistic samples while retaining the small distance between the actual and estimated distributions.

Recall the objective function of VAE proposed by Kingma & Welling (2014):

$$L_{\text{VAE}}(\theta, \phi) = -\mathbb{E}_{\mathbf{z} \sim q_\phi(\mathbf{z}|\mathbf{x})} \log p_\theta(\mathbf{x} \mid \mathbf{z}) + D_{\text{KL}}\left(q_\phi(\mathbf{z} \mid \mathbf{x}) \| p_\theta(\mathbf{z})\right). \tag{1}$$

Here, $p_\theta(\mathbf{x} \mid \mathbf{z})$ is the probabilistic decoder, $q_\phi(\mathbf{z} \mid \mathbf{x})$ is the recognition model, KLD is denoted by $D_{\text{KL}}(q_\phi(\mathbf{z} \mid \mathbf{x}) \| p_\theta(\mathbf{z} \mid \mathbf{x}))$ parameterized by the weights ($\theta$) and bias ($\phi$) of inference and generative models. As the incentive of $\beta$-VAE is to introduce the disentangling property, maximizing the probability of generating original data, and minimizing the distance between them, a constant $\delta$ is introduced in the objective VAE to formulate the approximate posterior distributions as below:

$$\max_{\phi, \theta} \mathbb{E}_{\boldsymbol{x} \sim \text{X}} \left[ \mathbb{E}_{q_\phi(z|\boldsymbol{x})} \left[ \log p_\theta(\boldsymbol{x} \mid z) \right] \right] \quad \text{such that} \quad D_{\text{KL}}\left(q_\phi(z \mid \boldsymbol{x}) \| p(z)\right) < \delta. \tag{2}$$

Rewriting the Equation in Lagrangian form and using the KKT conditions, Higgins et al. (2017) derive the following objective function:

$$\mathcal{L}_{\beta VAE}(\theta, \phi) = \mathbb{E}_{q_\phi(z|x)} \left[ \log p_\theta(x \mid z) \right] - \beta D_{\text{KL}}\left(q_\phi(z \mid x) \| p(z)\right), \tag{3}$$

Here, $\beta$ is the regularization coefficient that enforces the constraints to limit the capacity of the latent information z. When $\beta = 1$, we recover the original VAE. Increasing the value of $\beta > 1$ enforces the constraints to capture disentanglement. However, Hoffman et al. (2017) argue that with an implicit prior, optimizing the regularized ELBO is equivalent to performing variational expectation maximization (EM).

### 2.2 FACTORVAE

Disentangling by Factorising or FactorVAE is another modification of $\beta$-VAE proposed by Kim & Mnih (2018). FactorVAE emphasizes the trade-off between disentanglement and reconstruction quality. The authors primarily focused on the objective function of the VAE and $\beta$-VAE. The authors propose a new loss function to mitigate the loss of information that arise while penalizing both the mutual information and the KLD to enforce disentangled latent factors.

According to Hoffman & Johnson (2016) and Makhzani & Frey (2017), the objective function of $\beta$-VAE can be further extended into:

$$\mathbb{E}_{p_{\text{data}}(x)}[KL(q(z \mid x) \| p(z))] = I(x; z) + KL(q(z) \| p(z)), \tag{4}$$

Here, $I(x; z)$ is the mutual information between $x$ and $z$ under the joint distribution $p_{\text{data}}(x)q(z \mid x)$. FactorVAE learns the second term of $KL(q(z)\|p(z))$ and resolved the aforementioned issues by introducing total correlation penalty and density-ratio trick to approximate the distribution $\bar{q}(z)$ generated by $d$ samples from $q(z)$. The loss function of the FactorVAE is as follows:

$$
\mathbb{E}_{q\left(z|x^{(i)}\right)}\left[\log p\left(x^{(i)} \mid z\right)\right] - KL\left(q\left(z \mid x^{(i)}\right)\|p(z)\right) \\ - \gamma KL(q(z)\|q(z)) \tag{5}
$$

### 2.3 THE PRINCIPLE OF TOTAL CORRELATION EXPLANATION (CORED)

Gao et al. (2019) introduced CorEx to mitigate the problem of learning disentangled and interpretable representations in a purely information-theoretic way. In general, for VAE, we assume a generative model where $\mathbf{x}$ is a function of a latent variable $\mathbf{z}$, and afterward maximize the $\log$ likelihood of $\mathbf{x}$. On the other hand, CorEx follows the reverse process where $\mathbf{z}$ is a stochastic function of $\mathbf{x}$ parameterized by $\theta$, i.e., $p_\theta(\mathbf{z} \mid \mathbf{x})$, and seek to estimate the joint distribution $p_\theta(\mathbf{x}, \mathbf{z}) = p_\theta(\mathbf{z} \mid \mathbf{x})p(\mathbf{x})$. The underlying true data distribution maximizes the following objective:

$$
\mathcal{L}(\theta; \mathbf{x}) = \underbrace{TC_\theta(\mathbf{x}; \mathbf{z})}_{\text{informativeness}} - \underbrace{TC_\theta(\mathbf{z})}_{\text{(dis)entanglement}} \\ = TC(\mathbf{x}) - TC_\theta(\mathbf{x} \mid \mathbf{z}) - TC_\theta(\mathbf{z}). \tag{6}
$$

Recall the definition of the total correlation (TC) in terms of entropy $H(\mathbf{x})$ (Studený & Vejnarová, 1998):

$$
TC(\mathbf{x}) = \sum_{i=1}^{d} H(\mathbf{x}_i) - H(\mathbf{x}) = D_{KL}\left(p(\mathbf{x})\|\prod_{i=1}^{d} p(\mathbf{x}_i)\right). \tag{7}
$$

By non-negativity of TC, Equation 6 naturally forms variational lower bound $TC(\mathrm{x})$ to the CorEx objective, i.e., $TC(\mathrm{x}) \geq \mathcal{L}(\theta; \mathrm{x})$ for any $\theta$. Equation 6 can be rewritten in terms of mutual information $I(\mathrm{x} : \mathrm{z}) = H(\mathrm{x}) - H(\mathrm{x} \mid \mathrm{z}) = H(\mathrm{z}) - H(\mathrm{z} \mid \mathrm{x})$. Further constraining the search space $p_\theta(\mathbf{z} \mid \mathbf{x})$ to have the factorized form $p_\theta(\mathbf{z} \mid \mathbf{x}) = \prod_{i=1}^{m} p_\theta(\mathbf{z}_i \mid \mathbf{x})$ and the mutual information terms can be bounded by approximating the conditional distributions $p_\theta(\mathbf{x}_j \mid \mathbf{z})$ and $p_\theta(\mathbf{z}_j \mid \mathbf{x})$. Finally, we can further rewrite and derive the lower bound for the objective function:

$$
\mathcal{L}(\theta; \mathrm{x}) = \sum_{i=1}^{d} I_\theta(\mathrm{x}_i : \mathrm{z}) - \sum_{i=1}^{m} I_\theta(\mathrm{z}_i : \mathrm{x}) \\ \geq \left(\sum_{i=1}^{d} H(\mathrm{x}_i)\right) + E_{p_\theta(\mathrm{x},\mathrm{z})}\left(\log \underbrace{q_\phi(\mathrm{x} \mid \mathrm{z})}_{\text{decoder}}\right) \\ - D_{KL}(\underbrace{p_\theta(\mathrm{z} \mid \mathrm{x})}_{\text{encoder}}\|r_\alpha(\mathrm{z})). \tag{8}
$$

### 2.4 TOTAL CORRELATION VARIATIONAL AUTOENCODER ($\beta$-TCVAE)

Chen et al. (2018) proposed disentanglement in their learned representations by adjusting the functional structure of the ELBO objective. The authors argued that each dimension of a disentangled representation should be able to represent a different factor of variation in the data and be changed independently of the other dimensions. $\beta$-TCVAE modifies the originally proposed ELBO objective by Higgins et al. (2017) forcing the algorithm to learn representations without explicitly making restrictions or reduction to the latent space. Recall the ELBO objective function (Equation 3) of $\beta$-VAE:

$$\mathcal{L}_{\beta VAE}(\theta, \phi) = \mathbb{E}_{q_\phi(z|x)} \left[ \log p_\theta(x \mid z) \right] - \beta D_{\mathrm{KL}} \left( q_\phi(z \mid x) \| p(z) \right) \tag{9}$$

To introduce TC and disentanglement into the original $\beta$-VAE, Chen et al. decomposed the original KLD into **Index-Code MI, Total Correlation** and **Dimension-wise KL** terms. Furthermore, in the ELBO TC-Decomposition, each training samples are identified with a unique index $\mathbf{n}$ and a uniform random variable that refers to the aggregated posterior as $q(z) = \sum_{n=1}^{N} q(z \mid n) p(n)$ and can be denoted as:

$$\mathbb{E}_{p(n)}[\mathrm{KL}(q(z \mid n) \| p(z))] = \mathrm{KL}(q(z,n) \| q(z)p(n)) + \mathrm{KL}\left( q(z) \| \prod_j q(z_j) \right)$$
$$+ \sum_j \mathrm{KL}\left( q(z_j) \| p(z_j) \right) \tag{10}$$

Finally, with a set of latent variables $z_j$, with known factors $v_k$, the authors introduced a disentanglement measuring metric called mutual information gap (MIG) and defined in terms of empirical mutual information $I_n(z_j; v_k)$:

$$\frac{1}{K} \sum_{k=1}^{K} \frac{1}{H(v_k)} \left( I_n \left( z_{j(k)}; v_k \right) - \max_{j \neq j(k)} I_n \left( z_j; v_k \right) \right) \tag{11}$$

Here, $j^{(k)} = \mathrm{argmax}_j \, I_n(z_j; v_k)$ and $K$ is the number of known factors under $v_k$.

## 3 Disentangled Conditional Variational Autoencoder (dCVAE)

Our approach builds on CorEx and models known sources of variation in the data, in a manner similar to Conditional Variational Autoencoder (CVAE) Pol et al. (2019). In what follows, we will represent this known source of variation using the variable $C$. In the experiment below, $C$ is discrete and represents the class of each image. Modifying Equation 6 to incorporate $C$, we get

$$\mathcal{L}(\theta; x, c) = TC_\theta(x \mid c) - TC_\theta(x \mid z, c) - TC_\theta(z \mid c). \tag{12}$$

Recall that the first two terms measure the amount of correlation explained by $z$, and by maximizing it, we maximize the informativeness of the latent representation. The third term measures the correlation between the components of $z$, and by minimizing it, we maximize the disentanglement between the latent dimensions.

Using Mutual Information Theory (Studenỳ & Vejnarová, 1998), we can define the conditional differential entropy of $H(x)$ given $c$ and interpret mutual information as a reduction in uncertainty after conditioning:

$$I(x : z \mid c) = H(x \mid c) + H(z \mid c) - H(x, z \mid c)$$
$$I(x : z \mid c) = H(x \mid c) - H(x \mid z, c) = H(z \mid c) - H(z \mid x, c). \tag{13}$$

We can now rewrite Equation 12 using derived mutual information theory from Equation 13:

$$\mathcal{L}(\theta; x, c) = \sum_{j=1}^{p} I(x_j : z \mid c) - \sum_{j=1}^{d} I(z_j : x \mid c). \tag{14}$$

Now, consider the KLD between $p_\theta(\mathbf{x} \mid \mathbf{z}, c)$ and an approximating distribution $q_\phi(\mathbf{x} \mid \mathbf{z}, c)$. In terms of expectations with respect to the joint distribution $p_\theta(\mathbf{x}, \mathbf{z} \mid c)$, we can write:

$$-H(x \mid z, c) = E(\log p_\theta(x \mid z, c)) \geq E(\log q_\phi(x \mid z, c)). \tag{15}$$

Combing Equation 14 and 15 and assuming an approximating distribution $r_\alpha(z_j \mid c)$ for $p_\theta(z_j \mid c)$, we obtain two inequalities:

$$I\left(x_j : z \mid c\right) = H\left(x_j \mid c\right) - H\left(x_j \mid z, c\right) \geq H\left(x_j \mid c\right) + E(\log q_\phi(x \mid z, c)), \quad (16)$$

$$I\left(z_j : x \mid c\right) = D_{KL}\left(p_\theta\left(z_j \mid x, c\right) \| r_\alpha\left(z_j \mid c\right)\right). \quad (17)$$

Combining these bounds, we finally derive a lower bound for the objective function for dCVAE:

$$\mathcal{L}(\theta; x, c) \geq \sum_{j=1}^{p} H\left(x_j \mid c\right) + E\left(\log q_\phi(x \mid z, c)\right) - \sum_{j=1}^{d} D_{KL}\left(p_{(z_j|x,c)\|r(z_j|c)}\right). \quad (18)$$

Equation 18 illustrates the lower bound objective function of dCVAE where $q_\phi(\mathbf{x} \mid \mathbf{z}, c)$ is the generative model or decoder and $p_\theta\left(\mathbf{z}_j \mid \mathbf{x}, c\right)$ is the recognition model or encoder.

## 4 EXPERIMENTS

In the experiments below, we compare our dCVAE method to five baseline methods: VAE, CVAE, $\beta$-VAE, Factor-VAE, and RFVAE. The first two methods were selected as well-known baselines that do not explicitly enforce disentanglement; on the other hand, the latter three methods seek to achieve a disentangled representation of the data.

### 4.1 DATASETS

We evaluate dCVAE and other baseline models on the following four datasets. Three datasets (MNIST (Deng, 2012), Fashion-MNIST (Xiao et al., 2017), KMNIST (Clanuwat et al., 2018)) are trained for UAD. The fourth dataset (EMNIST (Cohen et al., 2017)) is used for testing on a real-world dataset to assess overall performance. A more detailed description of these datasets follows:

- **MNIST and Fashion-MNIST (FMNIST)** Firstly, we apply all models to two benchmark datasets, MNIST and Fashion-MNIST, for a fair comparison with other baseline methods. We used 10 classes with 60000 and 10000 training and testing samples for both datasets with $28 \times 28 \times 1$ pixels channel.

- **KMNIST** Secondly, we applied the same training process to another complex dataset, Kuzushiji-MNIST or KMNIST. KMNIST is a drop-in replacement for the MNIST dataset, a Japanese cursive writing style. KMNIST contains similar 10 classes with 60000 and 10000 training and testing samples with $28 \times 28 \times 1$ pixels channel.

- **EMNIST** Finally, all models are tested on the Extending MNIST or EMNIST Dataset. Using all 62 classes (digit 0-9, letters uppercase A-Z and lowercase a-z) with 700000 and 80000 training and testing samples with $28 \times 28 \times 1$ pixels channels. The dataset was processed from NIST Special Database 19 Grother (1995) and contained handwritten digits and characters collected from over 500 writers.

### 4.2 RECONSTRUCTION ERROR AND ANOMALY SCORE

Leveraging methods for the discriminator as the anomaly score and drawing separation between normal and anomalous data is challenging for the divergent architectures of autoencoders. Depending on the task the architecture is trained for, the discriminator varies greatly. In general, the UAD methods utilize reconstruction error (Baur et al., 2018), distribution-based error (Goldstein & Uchida, 2016), and density-based error (Kiran et al., 2018) scores to distinguish normal and anomalous data. Formally, for each input $x$, a test input $\widehat{x}_l$ is considered to be anomalous if reconstruction error or Anomaly score($\mathcal{A}$) is greater than the minimum threshold value and denoted as follows:

$$\mathcal{A}(\hat{x}) = \|x - \mathrm{D}(\mathrm{G}(\hat{x}))\|_2. \quad (19)$$

### 4.3 PERFORMANCE METRICS

One of the challenges of measuring the performance of disentanglement is to apply appropriate metrics based on the nature of the dataset, not of latent factors or dimensions in the latent space. Therefore, considering the different model architectures and datasets, we first measure the performance using Numerical AUC Score, reconstruction error ($\mathcal{A}$), and negative ELBO score ($\mathcal{E}$). These metrics provide a quantifiable method of accuracy, while also measuring the disentanglement among the latent factors.

We also measure performance qualitatively by visualizing the latent space and the 2D-manifold. Both allow us to visualize the orthogonality between latent features and demonstrate the accuracy of the models to handle reduced latent variables and the ability to reconstruct samples.

### 4.4 MODEL CONFIGURATION

A fixed set of hyper-parameters are chosen to formulate a similar platform for all models and identify the computational cost and reproducibility of the models. Although baseline models that we chose, $\beta$-VAE, FactorVAE, RFVAE are highly sensitive to hyper-parameters tuning, the hyper-parameters throughout the experiment are kept consistent to observe how the models perform under similar values. A minimal 50 epochs are used to train the datasets. For MNIST, FMNIST, and KMNIST the batch size is kept to 64, with primary and secondary learning rates as $\alpha = 10^{-5}$ and $\alpha = 10^{-3}$ respectively. However, for the EMNIST dataset, the batch size increased to 128, and learning rates as $\alpha = 10^{-6}$ and $\alpha = 10^{-5}$.

## 5 RESULTS AND DISCUSSION

In this section, we evaluate the results of dCVAE and other baseline methods on the downstream task of anomaly detection. A considerable volume of results was produced from our exhaustive evaluation. However, accounting for limitations of space here, we elected to focus on the results from EMNIST and KMNIST datasets in the main text. The remaining results (MNIST and FMNIST) are presented as Supplementary Material.

We show the results of our evaluation in three stages: firstly, using sample reconstruction and the negative ELBO score ($\mathcal{E}$) with reconstruction error $\mathcal{A}$, we evaluate and compare the disentanglement ability of dCVAE with baseline architectures. Secondly, we use the UMAP algorithm (Sainburg et al., 2021) to reduce dimensions and visualize both latent representation, as well as interpolation of the 2D-manifold to distinguish the TC by comparing information loss and effects of modeling known sources of variation. Finally, we present AUC scores and training time to summarize the overall accuracy of the experimented methods.

We evaluate the quality of disentanglement by considering explicit separation of $\mathcal{A}$ between normal and anomalous data and minimization of $\mathcal{E}$. A better disentanglement is achieved when:

(a) A higher reconstruction error $\mathcal{A}$ for anomalous sample and lower reconstruction error $\mathcal{A}$ for normal sample is obtained and

(b) $\mathcal{E}$ is minimized by enforcing regularization that either minimizes the negative ELBO decomposition $D_{KL}\left(p_{(z_j|x,c)\|r(z_j|c)}\right)$ or regularizes the approximate posterior $q_\phi(\mathbf{z} \mid \mathbf{x})$.

A clear boundary in terms of learning efficient disentanglement between dCVAE and baseline methods can be observed from both EMNIST (Figure 1) and KMNIST (Figure 2) reconstruction. The first row corresponds to anomalous reconstruction and the second row shows normal sample reconstruction. Both $\mathcal{E}$ and $\mathcal{A}$ score suggests that dCVAE captures more independent factors and identifies anomalous and normal samples efficiently. This observation strongly justifies one of our primary claims, namely that dCVAE incorporates the disentanglement learning through enforcing TC and restrict independent latent variables to prioritize the minimization of the divergence. The other disentanglement methods presented here either only emphasize TC (indicated by the dependence between random variables) or introduce $\beta$ (weighing the prior enforcement term), which limits the ability to learn randomness in a case when the hyperparameters are not tuned for certain dimensions.

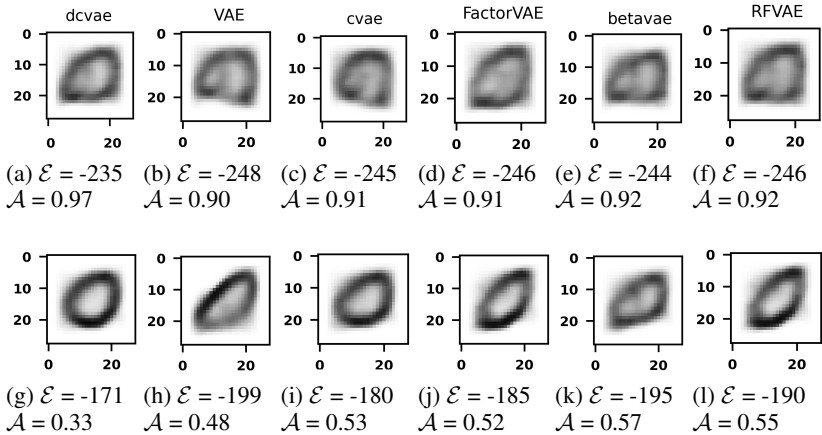

Figure 1: Reconstruction for digit zero (0) and the capital letter O. Here, $\mathcal{E}$ refers to Negative ELBO score and $\mathcal{A}$ is the reconstruction error or anomaly score. Only dCVAE and FactorVAE show steady improvement for both types of reconstruction. All the other methods misclassify the samples. Moreover, we can observe higher reconstruction error and ELBO scores compared to MNIST (Figure A1) and FMNIST (Figure A2).

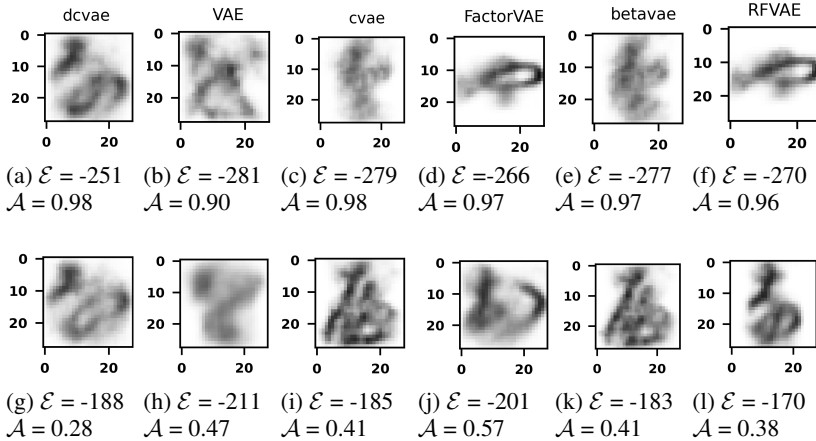

Figure 2: In KMNIST dataset, without dCVAE, all other methods fail to classify both anomalous and normal samples. Reconstruction scores suggest FactorVAE, VAE almost fail to distinguish normal and anomalous observations. Since the stroke of the samples are similar in this dataset, methods that only emphasize disentanglement or empirical approximation lose more information in latent variable resulting in false anomaly detection.

The second observation is drawn using latent representation (Figure 3) and 2D-manifold embeddings (Figure 4 and 5). Through this experiment, we observe the effect of modeling using a known source of variation (i.e. introducing conditional variable $C$ into the objective function) and minimizing information loss through multivariate mutual information theory (i.e. decomposition of TC). We can observe clear similarities between KLD loss and modeling with known score of variance in a reduced latent space. Due to enforced divergence loss, the plot of VAE and $\beta$-VAE are noticeably different from other architectures. Feature space is more compact for VAE, $\beta$-VAE, and we can see the cluster of the different classes are not well separated. However, conditioning the generative function (encoder) of CVAE and dCVAE provides the leverage to construct higher feature space and retain more accurate information in 2D-manifold (EMNIST, Figure 4; and KMNIST, Figure 5). Furthermore, TC reduces the correlation among disentanglement degrees when a specific feature is

learned (shape, strokes, color, boundaries). Such classes can be observed to cluster together and the other gets scattered with higher feature space (Figure 3). Compared to other methods, it is evident that dCVAE maintain consistent latent space and create separate clusters more accurately. This indicates that more disentangled variables are captured, and they retain more information through conditioning the generative model by minimizing the ELBO $D_{KL}\left(p_{(z_j|x,c)}\|r(z_j|c)\right)$.

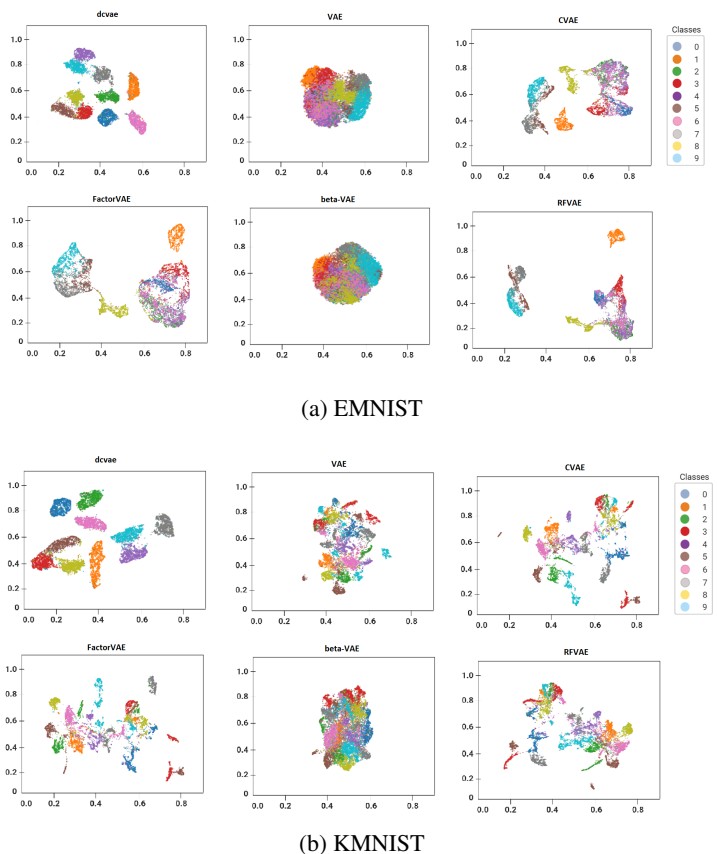

Figure 3: Latent Representation of EMNIST and KMNIST

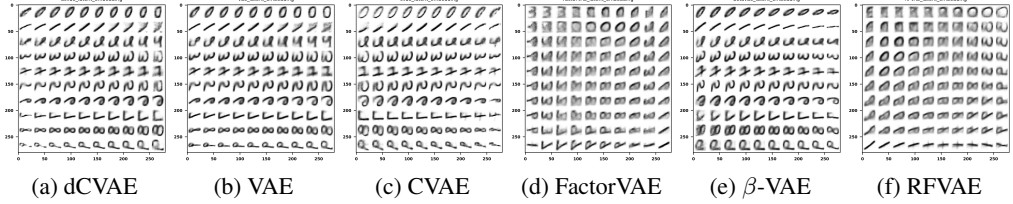

Figure 4: Manifold Embeddings (EMNIST)

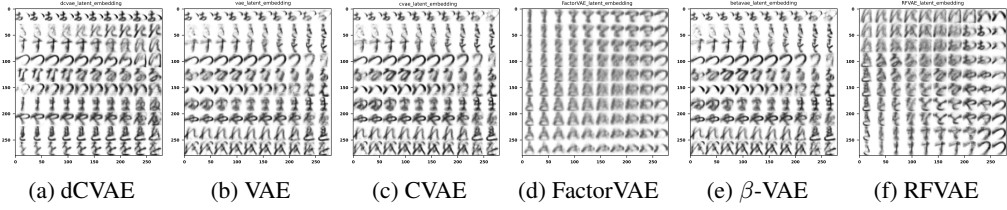

Figure 5: Manifold Embeddings (KMNIST)

Finally, Table 1 illustrates the results of model evaluation through AUC score and training time. dCVAE outperforms other methods in terms of AUC score. However, for larger divergent datasets like KMNIST and EMNIST, VAE shows lower training time compared to dCVAE. Since VAE only optimizes the negative log-likelihood, reconstruction loss and prior enforcement term, the training takes fewer latent variables to regularize, resulting in less training time. Nevertheless, compared to methods that incorporate TC (e.g. FactorVAE and RFVAE) or a constraint on the posterior ($\beta$-VAE), our proposed dCVAE scales to all larger datasets with higher classification accuracy.

Table 1: Evaluation metrics score

| Model | MNIST | | FMNIST | | EMNIST | | KMNIST | |
|---|---|---|---|---|---|---|---|---|
| | AUC | Training Time (min) | AUC | Training Time (min) | AUC | Training Time (min) | AUC | Training Time (min) |
| dCVAE | **88.31** | 37 | **88.63** | 44 | **78.98** | 102 | **61.02** | 95 |
| VAE | 88.21 | 37 | 84.12 | 39 | 67.23 | 92 | 51.13 | 78 |
| CVAE | 87.57 | 43 | 83.31 | 48 | 66.01 | 117 | 42.35 | 104 |
| FactorVAE | 87.11 | 53 | 82.78 | 50 | 62.91 | 138 | 49.23 | 117 |
| $\beta$-VAE | 85.31 | 51 | 82.31 | 53 | 65.12 | 123 | 50.01 | 119 |
| RFVAE | 85.31 | 55 | 81.11 | 57 | 55.03 | 130 | 49.51 | 132 |

The only trade-offs in our proposed method seem to occur when minimizing the negative ELBO loss. In certain conditions, dCVAE reaches a lower reconstruction loss (anomalous sample) yet minimizes the negative ELBO score (Figure 3, 4). In general, negative ELBO loss should illustrate symmetrical change with reconstruction error. Such inconsistency could lead to a significant drop in the classification accuracy, thus leading to a false anomaly detection result.

## 6 CONCLUSION

In this research, we present a novel generative variational model dCVAE, to improve the unsupervised anomaly detection task through disentanglement learning, TC loss, and minimizing trade-offs between reconstruction loss and reconstruction quality. Introducing a conditional variable to mitigate the loss of information effectively captures more disentangled features and produces more accurate reconstructions. Such architecture could be used in a wider range of applications, including generating controlled image synthesis, efficient molecular design and generation, source separation for bio-signals and images, and conditional text generation. Future research direction includes investigating in the gap between the posterior and the prior distribution, resolving the trade-offs between loss function and reconstruction, and inspect dCVAE using different disentanglement metrics.

### REPRODUCIBILITY STATEMENT

In this research, we carefully considered reproducibility in designing and conducting all experiments. In our supplemental texts, we have attached our source code. The experiments are designed independently to make the results reproducible. Image reconstruction and generation, 2D-Manifold embeddings, training time, and ELBO score calculation are performed separately from other downstream tasks like classification accuracy, reconstruction error, and latent representation. Furthermore, we used both TensorFlow and PyTorch frameworks to remove package dependencies. To remove the library dependencies and installation issues, virtual environment and package requirement files are also added. Finally, to make the results more accessible, we also provided randomly generated images with supplementary texts.

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

# A APPENDIX

Since we couldn't accommodate all results in our main paper, in this section we present results produced from MNIST and FMNIST datasets. The results are categorized into three sections: Reconstructions (A.1), Latent Representation (A.2), and 2D-Manifold embeddings (A.3).

## A.1 RECONSTRUCTION

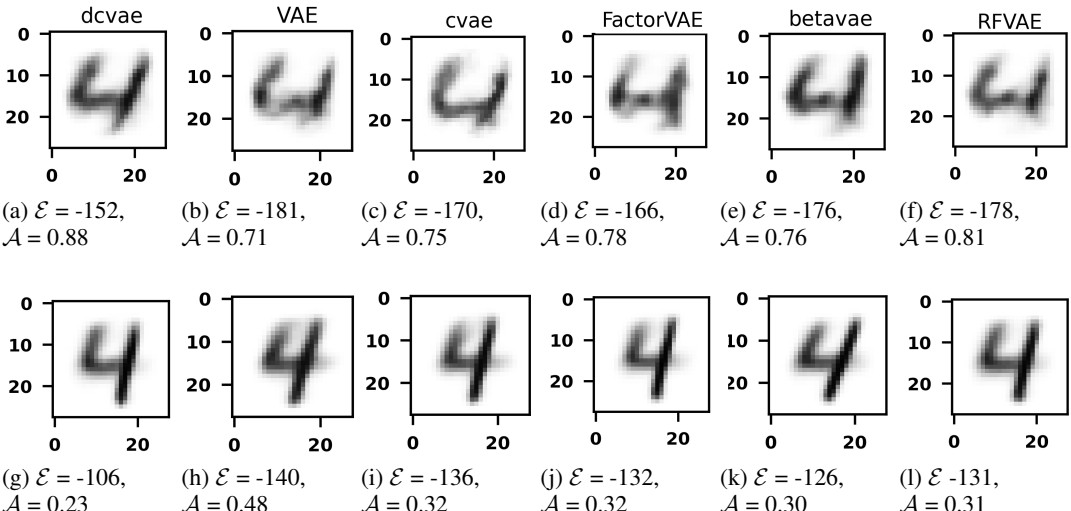

(a) $\mathcal{E}$ = -152, $\mathcal{A}$ = 0.88    (b) $\mathcal{E}$ = -181, $\mathcal{A}$ = 0.71    (c) $\mathcal{E}$ = -170, $\mathcal{A}$ = 0.75    (d) $\mathcal{E}$ = -166, $\mathcal{A}$ = 0.78    (e) $\mathcal{E}$ = -176, $\mathcal{A}$ = 0.76    (f) $\mathcal{E}$ = -178, $\mathcal{A}$ = 0.81

(g) $\mathcal{E}$ = -106, $\mathcal{A}$ = 0.23    (h) $\mathcal{E}$ = -140, $\mathcal{A}$ = 0.48    (i) $\mathcal{E}$ = -136, $\mathcal{A}$ = 0.32    (j) $\mathcal{E}$ = -132, $\mathcal{A}$ = 0.32    (k) $\mathcal{E}$ = -126, $\mathcal{A}$ = 0.30    (l) $\mathcal{E}$ -131, $\mathcal{A}$ = 0.31

Figure A1: The reconstruction from the MNIST dataset shows similar negative ELBO and reconstruction error ($\mathcal{A}$) values for CVAE, $\beta$-VAE, and RFVAE. our proposed model dCVAE performs best in terms of both reconstructing anomalous observation (first row) and normal observation (second row). We can observe a trade-off in FactorVAE with respect to $\beta$-VAE and RFVAE. FactorVAE performs better in reconstructing the anomalous observation whether as the $\beta$-VAE shows good performance in normal observations.

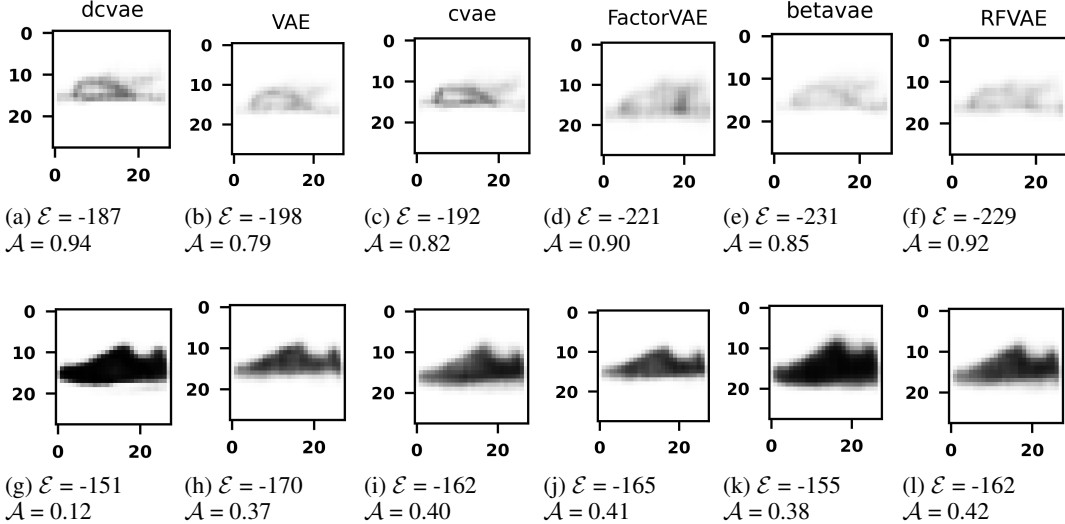

(a) $\mathcal{E}$ = -187 $\mathcal{A}$ = 0.94    (b) $\mathcal{E}$ = -198 $\mathcal{A}$ = 0.79    (c) $\mathcal{E}$ = -192 $\mathcal{A}$ = 0.82    (d) $\mathcal{E}$ = -221 $\mathcal{A}$ = 0.90    (e) $\mathcal{E}$ = -231 $\mathcal{A}$ = 0.85    (f) $\mathcal{E}$ = -229 $\mathcal{A}$ = 0.92

(g) $\mathcal{E}$ = -151 $\mathcal{A}$ = 0.12    (h) $\mathcal{E}$ = -170 $\mathcal{A}$ = 0.37    (i) $\mathcal{E}$ = -162 $\mathcal{A}$ = 0.40    (j) $\mathcal{E}$ = -165 $\mathcal{A}$ = 0.41    (k) $\mathcal{E}$ = -155 $\mathcal{A}$ = 0.38    (l) $\mathcal{E}$ = -162 $\mathcal{A}$ = 0.42

Figure A2: Similar to the MNIST dataset, the FMNIST illustrates similar trade-offs among FactorVAE, RFVAE, and $\beta$-VAE. However, for some samples, $\beta$-VAE mis-classifies the closely matched classes. dCVAE constrains the blurry reconstruction by enforcing conditions in the prior.

## A.2 LATENT SPACE VISUALIZATION

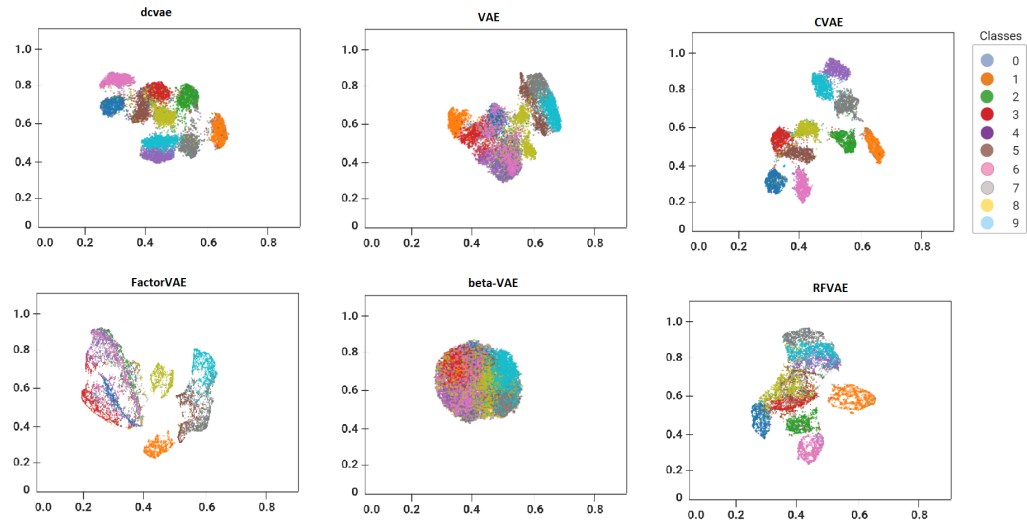

Figure A3: Latent Space Representation (MNIST)

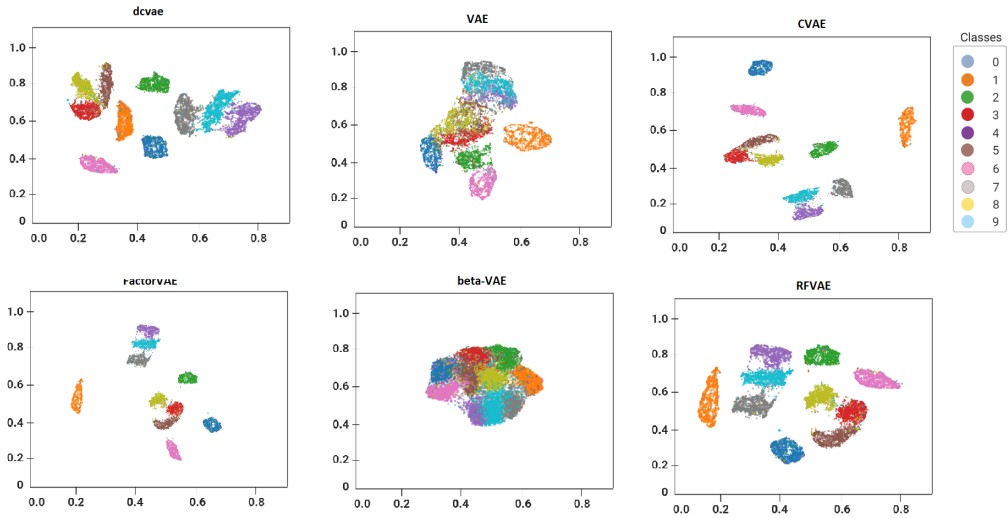

Figure A4: Latent Space Representation (FMNIST)

## A.3 LATENT MANIFOLD

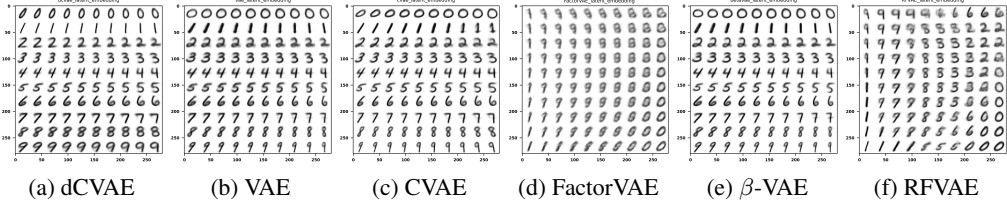

| (a) dCVAE | (b) VAE | (c) CVAE | (d) FactorVAE | (e) $\beta$-VAE | (f) RFVAE |

Figure A5: Latent Embeddings (MNIST)

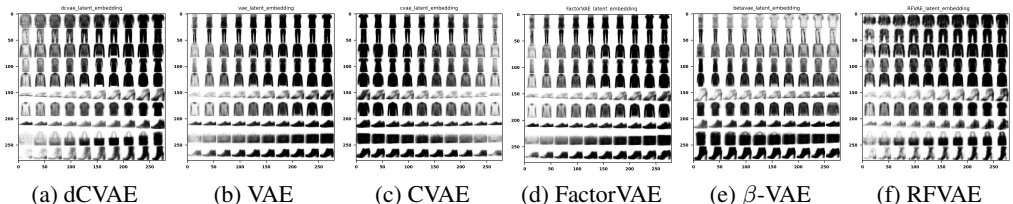

| (a) dCVAE | (b) VAE | (c) CVAE | (d) FactorVAE | (e) $\beta$-VAE | (f) RFVAE |

Figure A6: Latent Embeddings (FMNIST)

## A.4 RANDOM GENERATION

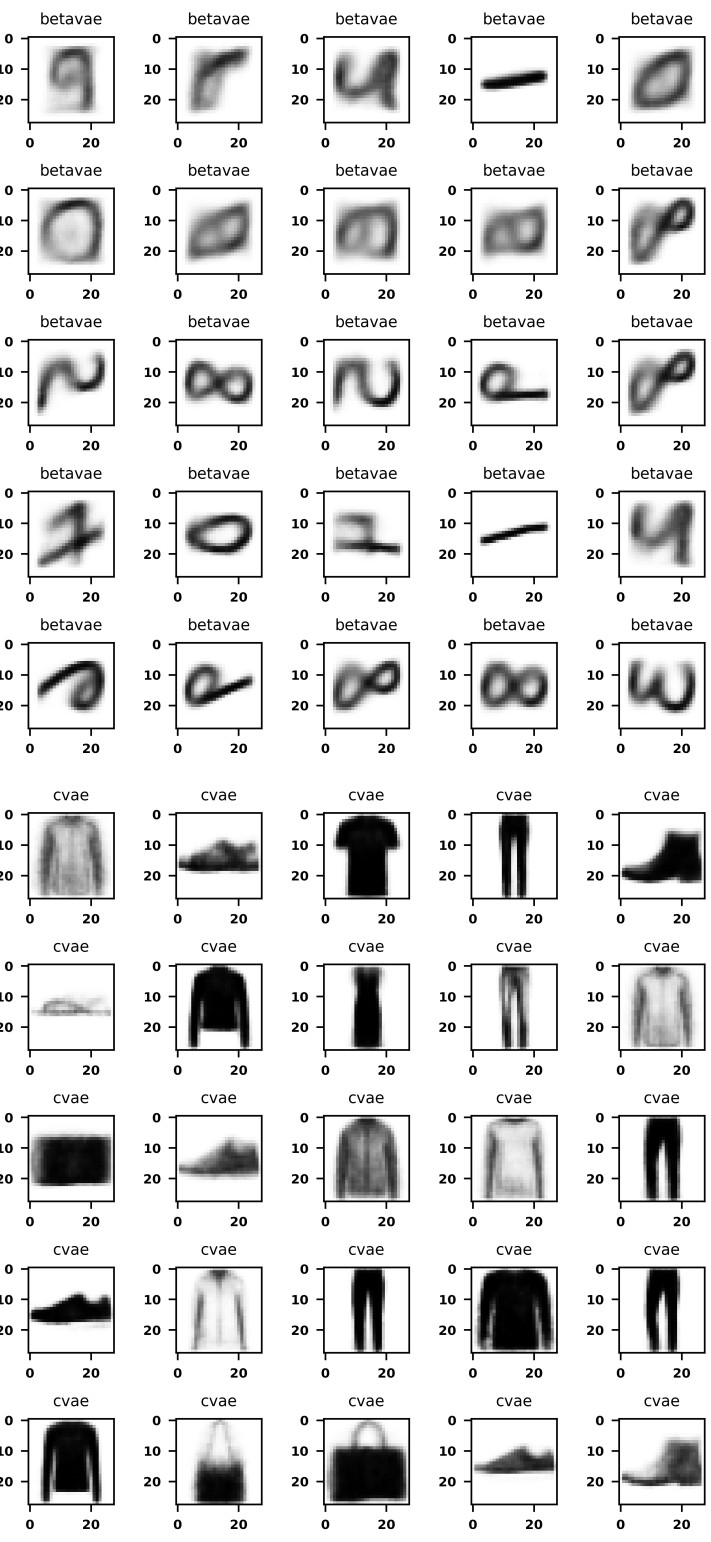

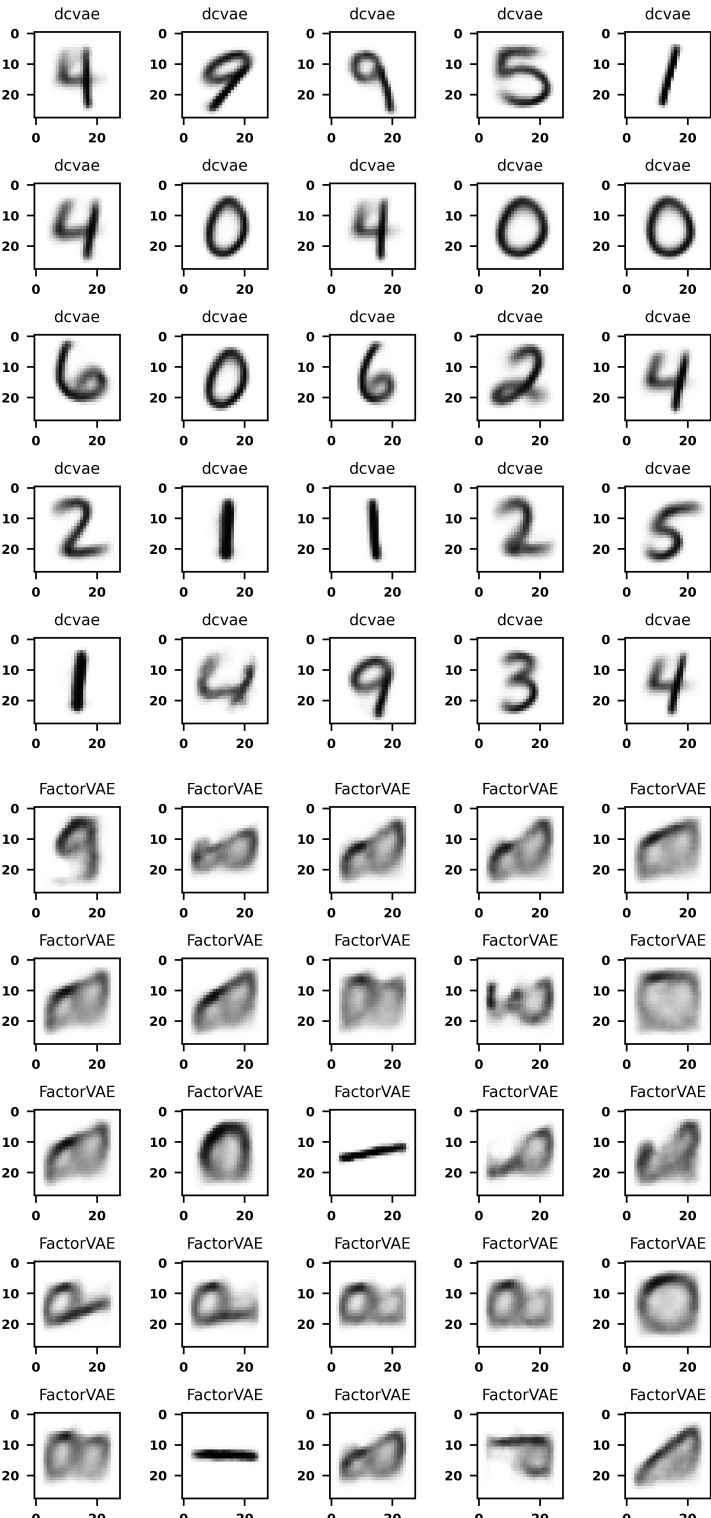

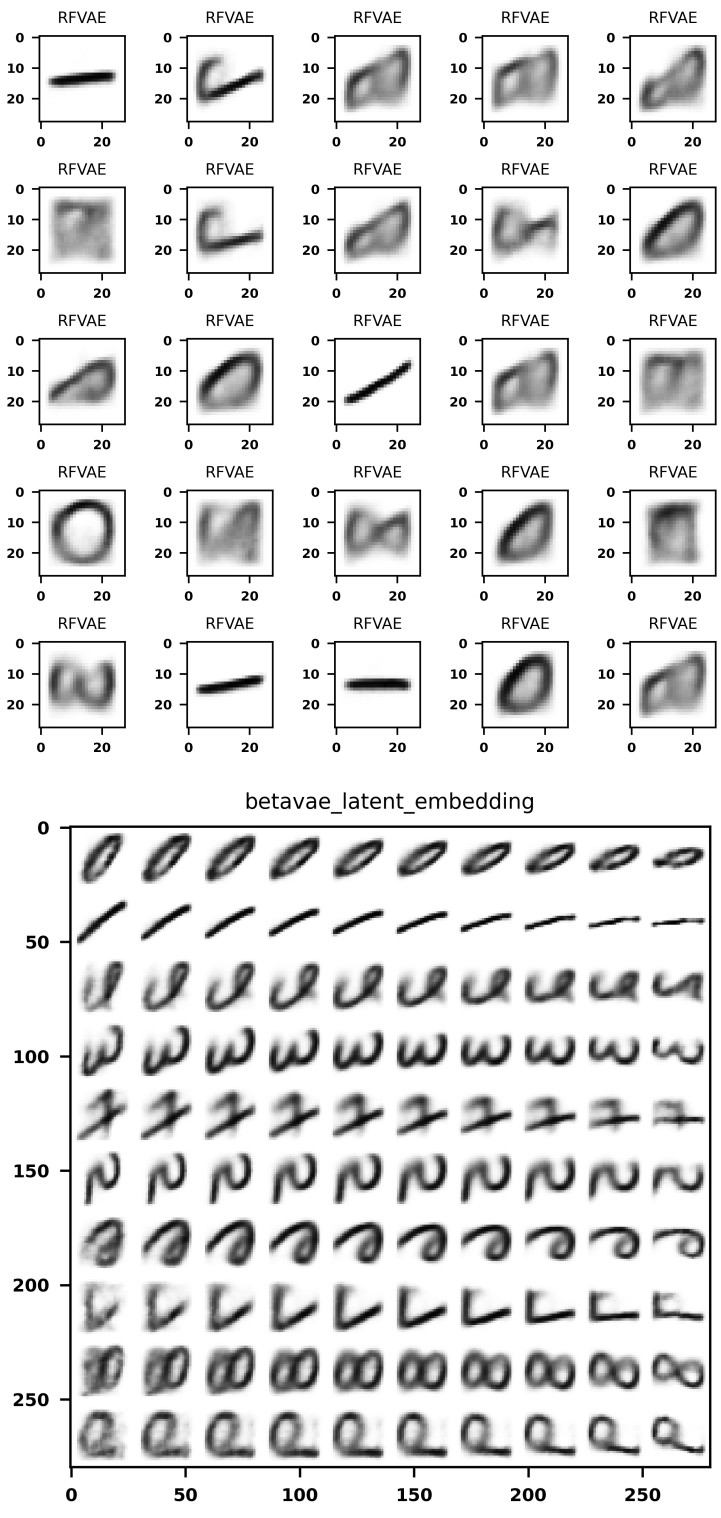

cvae_latent_embedding

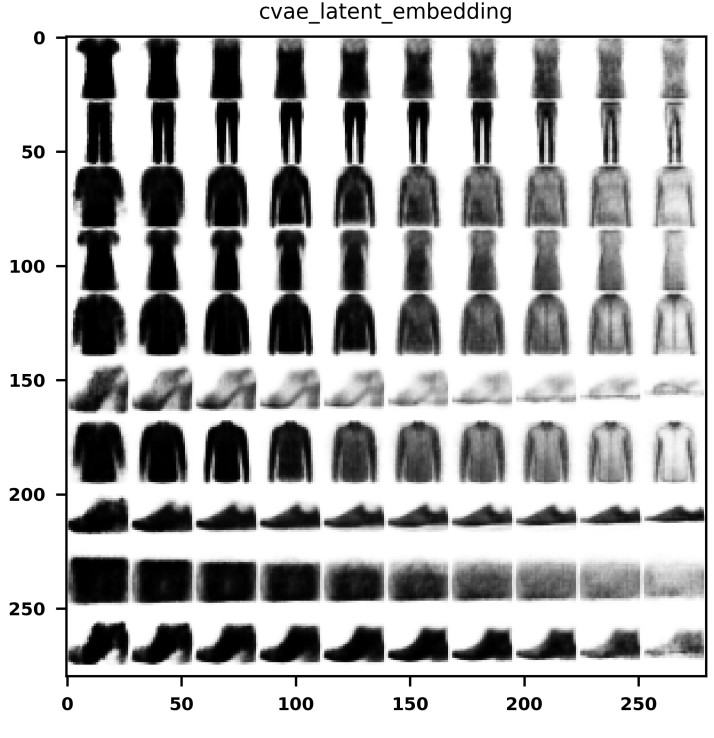

dcvae_latent_embedding

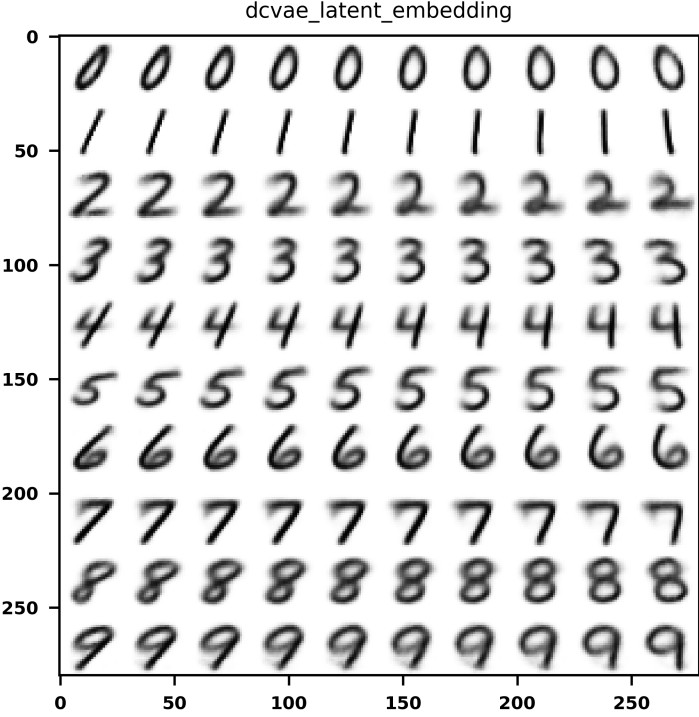

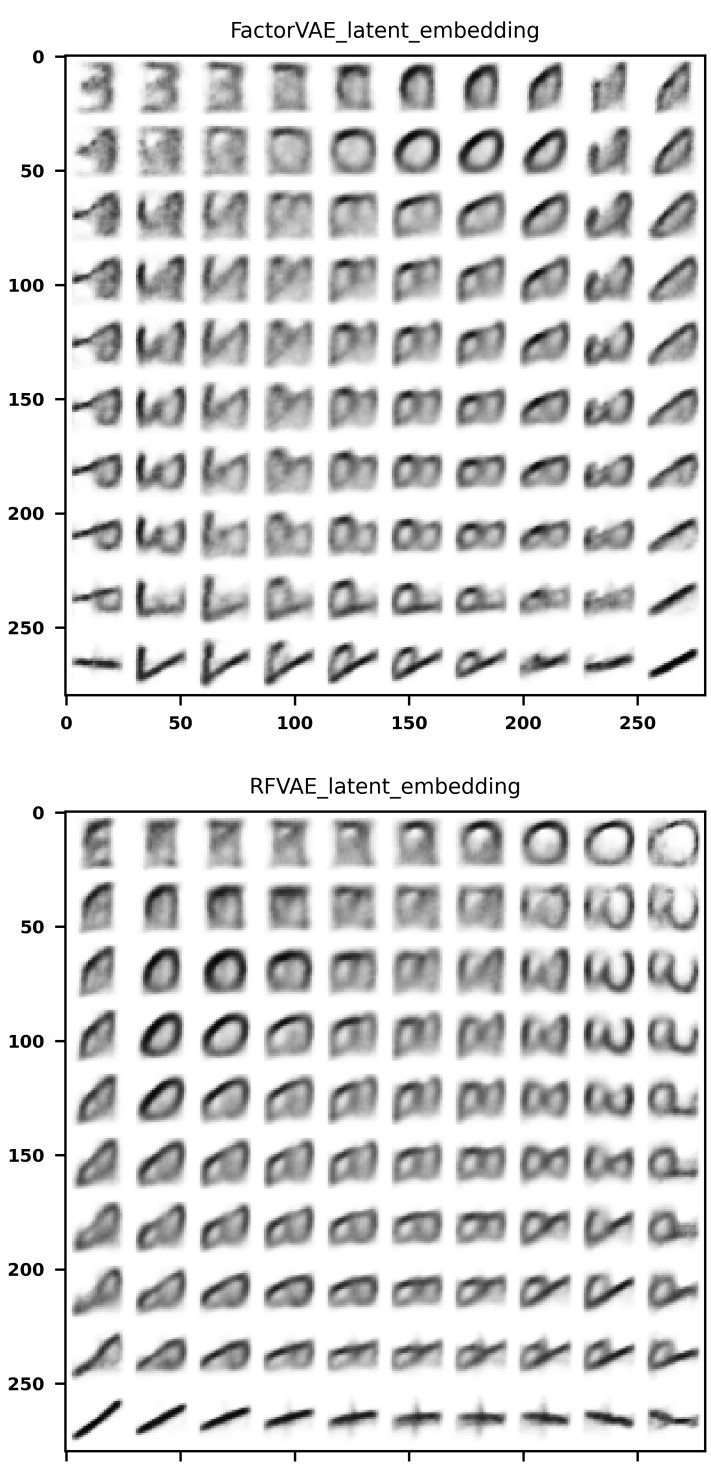

