# OpenReview forum: "Disentangled Conditional Variational Autoencoder for Unsupervised Anomaly Detection"
_ICLR.cc/2023/Conference — Submitted to ICLR 2023_

### Official Review · Reviewer_Pbh4 · 2022-10-20

**Confidence:** 4
**Correctness:** 3
**Technical Novelty And Significance:** 1
**Empirical Novelty And Significance:** 1
**Recommendation:** 3

**Clarity, Quality, Novelty And Reproducibility:**

**Clarity**

The first half of the paper is clear. The overview of related work is very detailed. The derivation of the proposed method and its lower bound is also clear.

The experiments are less clear, however. It seems EMNIST is considered as the anomaly and the other three are the training sets. However, Table 1 reports results for EMNIST too, which is confusing. Another point I find confusing is that the paper does not explain the use of conditioner $c$ in experiments. Are they labels? In that case, what is the $c$ for the test set?

**Quality**

There are some minor technical issues. For example, typo in eq (5). Also, there is no definition for the notation $r$ or $r_{\alpha}$.

**Novelty**

This is a major weakness of the paper. I do not see much novelty in methodology except that everything is now conditioned. The experiments also lack novelty: the method is only tested on several simple datasets with slightly increased AUC.

**Reproducibility**

The supplementary file includes the full code to reproduce the experiments. I went over the code but did not run the experiments.

**Strength And Weaknesses:**

The paper has a good overview on related work, including derivations of those baseline methods and building blocks of the proposed method. However, the paper has the following major weakness. First, in the proposed method, every term is in fact the same as CorEx except that they are conditioned on $c$. I do not see any novel ideas or insights from the proposed method. Second, the experiments are only ran on simple datasets based on MNIST and its variants. The paper does not conduct experiments on more complicated datasets, especially those reported in corresponding papers of the baseline methods. Furthermore, there is only one metric (reconstruction error) that has been evaluated. However, simple metrics sometimes fail to detect anomaly (Choi et al 2021).

**Summary Of The Paper:**

The paper presents disentangled conditional VAE for anomaly detection. The model and lower bound are based on a combination of total correlation and CVAE in the $\beta$-VAE framework. The paper then conducts experiments on several grey-scale image datasets. Specifically, the paper uses reconstruction error to detect outliers, similar to some prior work. Results show a higher AUC compared to those baselines.

**Summary Of The Review:**

Overall I think the paper does not have much contribution to methodology or experiments. Therefore, I believe it needs further improvement.

---

> ### Author Response · Authors · 2022-11-19
> **Response to Reviewer pbh4**
>
> Thank you for your review and comments; we have prepared a few points to clarify the observations:
>
> > ### " Second, the experiments are only ran on simple datasets based on MNIST and its variants. The paper does not conduct experiments on more complicated datasets, especially those reported in corresponding papers of the baseline methods. "
>
> &
>
> > ### "The experiments are less clear, however. It seems EMNIST is considered as the anomaly and the other three are the training sets. However, Table 1 reports results for EMNIST too, which is confusing."
>
> **Response:**
>  - We used the MNIST and Fashion-MNIST datasets as the primary benchmark for the baseline methods. Then, we designed to evaluate our models on EMNIST and KMNIST. Although they are a drop-in replacement for the MNIST dataset, to better understand our contribution in terms of efficiently utilizing the disentangled factors and minimizing information loss for UAD task, the following properties of the MNIST and KMNIST dataset makes this task heavily challenging: \enquote{In general, learning anomalous samples from the first two datasets is straightforward; however, EMNIST and KMNIST datasets contain small variations among normal and abnormal classes, sharp strokes, and lower distinguishable factors that result in severe challenges for UAD.} Moreover, the observations from latent representation and sample generation illustrate the highly diverse result that we believe our model showed significant improvement compared to baseline methods.
>
> > ### "Another point I find confusing is that the paper does not explain the use of conditioner $c$  in experiments. Are they labels? In that case, what is the $c$ for the test set? "
>
> **Response:**
>  - We adopted the conditional VAE architecture proposed by pol et al. (2019) and utilized the idea of controlled generation. For each sample data in the test dataset, we chose a random threshold to distinguish normal and anomalous data. Similarly to a conditional vector in CVAE, $C$ controls the threshold on that sample to generate controlled samples on either normal or anomalous reconstruction based on the attained threshold value.
>
>
> > ### "There are some minor technical issues. For example, typo in eq (5). "
>
> **Response:**
>  - In the updated version of our paper, the notation and the definition is included in the equation.

---

### Official Review · Reviewer_8jzN · 2022-10-25

**Confidence:** 4
**Correctness:** 3
**Technical Novelty And Significance:** 2
**Empirical Novelty And Significance:** 3
**Recommendation:** 3

**Clarity, Quality, Novelty And Reproducibility:**

The paper have very unclear notation that seems to be changing. The language / grammar in the paper is fine though.

**Strength And Weaknesses:**


Strengths:
- The overall idea seems interesting
- The results of Figure 3 and Table 1 look good

Major Weaknesses:
- It seems like the primary contribution of this paper is to condition previous work with a class variable c. Everything from equation 12 onwards follows the same exact things presented in Section 2, except with a conditioning on c. Furthermore, nothing is described on why this particular architecture is relevant or useful for anomaly detection.

- Why is CorEx not compared to in the experiments? Particularly because this work heavily depends on and builds off the model from CorEx.

Other Weaknesses:

- The notation is either inconsistent or extremely hard to follow. Making it difficult to review. Up until section 2.4 it seems that x is d dimensional and z is m dimensional and factorizable when conditioned on x. Although for some reason the index switches from i to j in the final sentence before equation 8. Do i and j represent different things? In section 2.4, j is used as the index and n is used to denote that there are N samples. Additionally k is used to denote the K factors v of z_j, which implies that each z_j has K factors v_k? Also what is total number of j (the sum and products are missing top values). In section 2.5, the j index is used, but now x is p dimensional and z is d dimensional? This confusion in notation makes it impossible to use the definition of TC in equation 7 with the decompositions in equation 13 to get equation 14. It also seems equation 8 is more useful in understanding the relationship between the TC for and the mutual information form of the loss function.

- Equation 16 is derived using 13 and 15, not 14 and 15?

- Below equation 7 you have I(x : z) instead of I(x ; z), Equation 13 and 14 are also using : instead of ;

- The objective function in equation 18 has no regularization / penalty term? What does it have to do with the various \beta or Factor models?

- The equations take up a lot of space and could be organized better so that more detail about the equations is described above / below it. Some equations definitely do not need to take multiples lines and do.

- There is a random l in \hat{x} above equation 19. Also what are x and \hat{x} paired or is the anomaly score over all x in the training set? This is not clear.

- Despite the text, in Figures 1 and 2, all methods have higher A for the top row (anomalous) versus the bottom row, which according to bullet a) of better disentanglement, is desired. Also all the \Epsilon values seems relatively close for all methods.

**Summary Of The Paper:**

This paper proposes a conditional VAE that uses total correlation to form their loss function. It builds off the CorEx work. They show many experimental results, some of which are promising.

**Summary Of The Review:**

Between the issues with clarity and the lack of comparison with the CorEx model in the experimental results, it is completely unclear whether there is significant improvement of the CorEx model. In general, this paper seems to rely heavily on previous work, with questionable amounts of novelty on its own.

---

> ### Author Response · Authors · 2022-11-19
> **Response to Reviewer 8jzN**
>
> Thank you for the comments and suggested corrections. We have addressed the issues as follows:
>
> > ### " It seems like the primary contribution of this paper is to condition previous work with a class variable c. Everything from equation 12 onwards follows the same exact things presented in Section 2, except with a conditioning on c. Furthermore, nothing is described on why this particular architecture is relevant or useful for anomaly detection. "
>
> **Response:**
> - In the original CVAE method, the conditioning variable $\mathbf{C}$ is introduced to control the generation of VAE. However, Both VAE and CVAE architecture lacks in addressing learning disentangled factors as well as addressing multivariate information theory for efficiently minimizing information loss. We took the aforementioned limitations and combined them into a new method that addresses the issue of disentangled learning, optimizing information loss, and improving the reconstruction ability (illustrated in section 3). Our main contribution is a generative modeling architecture that learns disentangled data representations while minimizing the loss of information and thus maintaining good reconstruction capabilities. We achieve this by modeling known sources of variation in a similar fashion as CVAE. Through our experimental design, we also illustrated how dCVAE shows an improvement after combining different architectures and addressing the capabilities simultaneously. We also added an ablation study to better illustrate the significance of dCVAE.
>
> > ### "Why is CorEx not compared to in the experiments? Particularly because this work heavily depends on and builds off the model from CorEx "
>
> **Response:**
> - Thank you for your observation of our selected methods. Although we referred to the baseline methods from prior work on disentangled learning and minimizing information loss, our primary task is to improve anomaly detection by combining these two techniques and achieve
> this by modeling known sources of variation, in a similar fashion as Conditional VAE. In the original paper of Gao et al. (2019), the authors proposed CorEx and the core applications on: Disentangling Latent Codes via Hierarchical VAE / Stacking CorEx on MNIST, Learning Interpretable Representations through Information Maximizing VAE / CorEx on CelebA, Generating Richer and More Realistic Images via CorEx. Since our dCVAE primarily contributes to unsupervised anomaly detection tasks by learning disentangled representations of the data while minimizing the loss of information and thus maintaining good reconstruction capabilities., we only chose the VAE extensions ($\beta$-VAE, CVAE, FactorVAE, RFVAE) that addresses results of anomaly detection to show how dCVAE improves this task and contricutes to the literature. However, we would consider accommodating more methods in the future for extracting diverse results.
>
>
> > ### " Other weakness - corrections on mathematical notations "
>
> **Response:**
>  - We carefully observed our equations 10-16 and updated corrections to your points 1-5.
>
>
> > ### " There is a random l in \hat{x} above equation 19. Also what are x and \hat{x} paired or is the anomaly score over all x in the training set? This is not clear. "
>
> **Response:**
> - To distinguish normal samples and anomalous samples, we referred to them as $x$ and $\hat{x}$, respectively, in the equation. If the test sample is anomalous, the anomaly score will be calculated in terms of $\hat{x}$, and for a normal instance, it will be $x$.
>
>
> > ### " Despite the text, in Figures 1 and 2, all methods have higher A for the top row (anomalous) versus the bottom row, which according to bullet a) of better disentanglement, is desired. Also all the \Epsilon values seems relatively close for all methods "
>
> **Response:**
> - To compare our results, we considered anomaly score and negative elbo minimization. Prior works in meta-priors (bengio et al. 2013), including disentanglement, Hierarchical organization of explanatory factors, heavily depend on the structure of minimizing information loss and elbo optimization. The baseline methods are heavily sensitive to the elbo score and their reconstruction quality. A small change in the elbo in a context where the disentangled factors are not equally learned shows the trade-off between "reconstruction error vs. reconstruction quality" that we actively pursued in our dCVAE method. Therefore we observed small deviations in $\mathcal{E}$ that resulted in causing changes (reconstruction and latent visualization) in anomaly detection performance.

---

### Official Review · Reviewer_HT5F · 2022-10-29

**Confidence:** 4
**Correctness:** 3
**Technical Novelty And Significance:** 2
**Empirical Novelty And Significance:** 2
**Recommendation:** 3

**Clarity, Quality, Novelty And Reproducibility:**

- Clarity: it is good to follow. The idea is straightforward.
- Quality: related works are high but the methodology section is not solid
- Novelty: it is weak and lack of major contribution

**Details Of Ethics Concerns:**

No ethics concerns

**Strength And Weaknesses:**

- Strength

1. Detail introduction about the related works, specifically the core techniques, i.e., beta-VAE, CVAE, and TC
2. The idea is simple to follow and the paper is well-written.

- Weaknesses

1. The novelty is weak.
2. Lack of ablation study on different components
3. The methodology section is not well explained with enough details and theoretical proof


**Summary Of The Paper:**

This paper focuses on addressing unsupervised anomaly detection via disentangled conditional VAE.  The new architecture combines three core components: beta-VAE, CVAE and the principle of TC.  The authors claim that the new method improves the disentanglement of latent features, and the ability to detect anomalies.  Multiple experiments are used to demonstrate the performance and improvement, specifically from the perspective of capturing disentangled features.

**Summary Of The Review:**

The paper seems to be close to the original CVAE method, which is not very new.  Combining three core methods is fine but we did not understand why the new ensemble works better than before, and why one is more important.  The authors should provide an ablation study to investigate the sensitivity of each component.

For experiments, the major datasets are MNIST-based data, which is a naive and easy case. It would be more strong if the authors can apply the method for large datasets typically used in AD/OOD detection tasks, e.g., CIFAR-10/100 or ImageNet.

In addition, the authors may put more effort into improving the new methodology section with more details and explanations. The related works are good enough but should be compressed if the space is not enough.

---

> ### Author Response · Authors · 2022-11-19
> **Response to Reviewer Reviewer HT5F**
>
> Thank you for your reviews and comments, we have prepared a few points to clarify the comments
>
> > #### "Lack of ablation study on different components."
>
> **Response:**
>  - We added an ablation study (Section 6) to address this concern. We divided our study into two parts: tuning parameters to see how the models perform under different conditions and generating samples using the optimized parameters. We also included Table 2 with the results and details of the ablation study.
>
> > #### "The methodology section is not well explained with enough details and theoretical proof."
>
> **Response:**
>  - Taking the concerns, here is a details explanation: For each figure from 1-5, we address the results connecting the theoretical background briefly mentioned in Section 2. We show the results of our evaluation in three stages: firstly, using sample reconstruction and the negative ELBO score ($\mathcal{E}$) with reconstruction error $\mathcal{A}$, we evaluate and compare the disentanglement ability of dCVAE with baseline architectures. Secondly, we use the UMAP algorithm to reduce dimensions and visualize both latent representations, as well as interpolation of the 2D-manifold to distinguish the TC by comparing information loss and effects of modeling known sources of variation. Finally, we present AUC scores and training time to summarize the overall accuracy of the experimented methods. We evaluate the quality of disentanglement by considering the explicit separation of $\mathcal{A}$ between normal and anomalous data and minimization of $\mathcal{E}$.
>
> > #### "The paper seems to be close to the original CVAE method, which is not very new. Combining three core methods is fine but we did not understand why the new ensemble works better than before, and why one is more important. The authors should provide an ablation study to investigate the sensitivity of each component."
>
> **Response:**
>  - In the original CVAE method, the conditioning variable $\mathbf{C}$ is introduced to control the generation of VAE. However, Both VAE and CVAE architecture lacks in addressing learning disentangled factors as well as addressing multivariate information theory for efficiently minimizing information loss. We took the aforementioned limitations and combined them into a new method that addresses the issue of disentangled learning, optimizing information loss, and improving the reconstruction ability. Our main contribution is a generative modeling architecture that learns disentangled data representations while minimizing the loss of information and thus maintaining good reconstruction capabilities. We achieve this by modeling known sources of variation in a similar fashion as CVAE. Through our experimental design, we also illustrated how dCVAE shows an improvement after combining different architectures and addressing the capabilities simultaneously. We also added an ablation study to better illustrate the significance of dCVAE.
>
>
> > #### "For experiments, the major datasets are MNIST-based data, which is a naive and easy case. It would be more strong if the authors can apply the method for large datasets typically used in AD/OOD detection tasks, e.g., CIFAR-10/100 or ImageNet."
>
> **Response:**
>  -  We used the MNIST and Fashion-MNIST datasets as the primary benchmark for the baseline methods. Then, we designed to evaluate our models on EMNIST and KMNIST. Although they are a drop-in replacement for the MNIST dataset, to better understand our contribution in terms of efficiently utilizing the disentangled factors and minimizing information loss for the UAD task, the following properties of the MNIST and KMNIST dataset makes this task heavily challenging: \enquote{In general, learning anomalous samples from the first two datasets is straightforward; however, EMNIST and KMNIST datasets contain small variations among normal and abnormal classes, sharp strokes, and lower distinguishable factors that result in severe challenges for UAD.} Moreover, the observations from latent representation and sample generation illustrate the highly diverse result that we believe our model showed significant improvement compared to baseline methods.
>
>
> > #### "In addition, the authors may put more effort into improving the new methodology section with more details and explanations. The related works are good enough but should be compressed if the space is not enough."
>
> **Response:**
>  - We have updated the related work section to shorten up few equations. In the first place, we derived each equation using the prior work, so the broad theoretical background is easy to follow. However, considering the comments, we merged the equations to facilitate the same idea in a concise manner.

---

### Official Review · Reviewer_DSrs · 2022-10-30

**Confidence:** 3
**Correctness:** 2
**Technical Novelty And Significance:** 2
**Empirical Novelty And Significance:** 2
**Recommendation:** 3

**Clarity, Quality, Novelty And Reproducibility:**

The work is highly incremental in nature and the empirical analysis is insufficient in its current form.

**Strength And Weaknesses:**

**Strengths**

- Paper is straightforward and relatively easy to follow.


**Weakness**

- The paper has this major flaw in its structure
a) Excessive emphasis on related work and prior art sections which is not necessary

- For reasons not known, the authors have conveniently ignored most prominent unsupervised deep anomaly baselines such as
  ### Bo Zong, Qi Song, Martin Renqiang Min, Wei Cheng, Cristian Lumezanu, Daeki Cho, and Haifeng Chen. 2018. Deep autoencoding gaussian mixture model for unsupervised anomaly detection. In International Conference on Learning Representations.

 ### Unsupervised Anomaly Detection with Adversarial Mirrored AutoEncoders. Somepelli et al. UAI 2021



- I also believe that anomaly detection frameworks need to benchmarked with results on metrics such as AUPR and not just indicate AUC improvements.


**Summary Of The Paper:**

In this paper, the authors proposed a framework for unsupervised anomaly detection which builds on top of existing prior art on disentangled variational autoencoders (VAE's). Authors do a deep dive on existing prior art techniques such as $\beta$ - VAE, Factor VAE and others by introducing a conditional variable to avoid loss of information and produce more accurate reconstructions. Authors present empirical results on several image datasets by demonstrating reconstruction score, accuracy and other important metrics for anomaly detection.

**Summary Of The Review:**

Unsupervised anomaly detection is an important problem, but new papers in this space need to be more novel as there is exists a significant body of work in this space (within both VAE's and GAN's). In addition, empirical evaluation should make sure the coverage is exhaustive to conclude if the results are significant. The current paper lacks most of these insights.

---

> ### Author Response · Authors · 2022-11-19
> **Response to Reviewer DSrs**
>
> We would like to thank you for your constructive review. We would like to address the following issues:
>
> > ####  "The paper has this major flaw in its structure a) Excessive emphasis on related work and prior art sections which is not necessary."
>
>  **Response:**
> - We have updated the related work section to shorten up few equations. In the first place, we derived each equation using the prior work, so the broad theoretical background is easy to follow. However, considering the comments, we merged the equations to facilitate the same idea in a concise manner.
>
> > #### "For reasons not known, the authors have conveniently ignored most prominent unsupervised deep anomaly baselines such as Bo Zong, Qi Song, Martin Renqiang Min, Wei Cheng, Cristian Lumezanu, Daeki Cho, and Haifeng Chen. 2018. Deep autoencoding gaussian mixture model for unsupervised anomaly detection. In International Conference on Learning Representations. Unsupervised Anomaly Detection with Adversarial Mirrored AutoEncoders. Somepelli et al. UAI 2021."
>
> **Response:**
> - We thank you for mentioning the extensive relevant works. To narrow down the literature review, we emphasized the methods closely related to disentangled learning, mutual information theory, and CorEx. Moreover, in our literature review, we also found the VAE extensions are more efficient for unsupervised anomaly detection and learning such features. Therefore, considering the scope of this study, we confined the models only to the extensions of VAE so that we could create a clear line between different baseline methods to compare our proposed dCVAE.
>
> > #### "I also believe that anomaly detection frameworks need to benchmarked with results on metrics such as AUPR and not just indicate AUC improvements"
> **Response:**
> - In our baseline comparison methods, the accuracy or performance is measured in different metrics and scales (e.g., disentangled metrics ($\beta$-VAE), One factor varied. (FactorVAE),  mutual information gap (MIG)). Moreover, the metrics use various measuring techniques (disentangled score, accuracy score on particular class) to assess the overall performance. Since we utilized those divergent models, we chose a mixture of quantitative and qualitative metrics to create a uniform baseline to evaluate our proposed model. We used the AUC score to measure the overall accuracy in anomaly detection, which is the main UAD task studied here. In prior works, especially for state-of-the-art AE models,  AUC is extensively used to summarize the classifier performance in a single number. Finally, about the benchmark datasets, we chose the MNIST and Fashion-MNIST datasets to compare them with current state-of-the-art methods and the KMNIST and EMNIST datasets to further evaluate the dCVAE's accuracy compared with baseline methods.

---

### Author Response · Authors · 2022-11-19
**Summary of Changes**

Dear Reviewers,

Thank you for taking the time to read our manuscript and provide precise feedback. Our individual responses can be found below, with the issues that have been addressed in the revised paper.

The manuscript has been updated with all of the suggested changes:

* Considering the space limitation, the related work section is compressed, and the equations are allocated in a precise manner.
* Equations and notations are addressed and updated based on the comments and concerns of the reviewers.
* In sec. 4, details on the dataset and experiments are refined.
* Modifications and additions to the results in Sec. 5,
* We added the ablation study in sec. 6 and pointers to new results in the Appendix,
* Miscellaneous comments from reviewers.

More details are available in the individual responses. Please let us know if you have any additional questions.

Sincerely,
Authors

---

### Decision · Program_Chairs · 2023-01-20

**Decision:**

Reject

**Justification For Why Not Higher Score:**

Four reviewers have consistent recommendations, i.e., "3: reject, not good enough". During the discussion phase, reviewers found that the responses from authors are not very convincing, and the major concern on novelty still remains.

**Justification For Why Not Lower Score:**

N/A

**Metareview: Summary, Strengths And Weaknesses:**

This paper presents a new architecture of generative auto-encoder for unsupervised anomaly detection. The proposed method leverages existing techniques such as $\beta$-VAE and CVAE, and employs a conditional variable for effective reconstruction. Experiments on benchmarks show promising results on anomaly detection.

Overall, this paper is well organized and clearly written. The authors have provided a comprehensive introduction of related work, and technical details of the proposed method are easy to follow.

However, this paper has some weaknesses, such as the limited novelty, insufficient experiments, missing baselines, and the lack of ablation studies. Although the authors provided responses to the comments, reviewers still concerned about the overall contribution of this paper, especially the limited novelty compared with existing work.

**Summary Of Ac-Reviewer Meeting:**

N/A